# Disulfiram/Copper Induces Immunogenic Cell Death and Enhances CD47 Blockade in Hepatocellular Carcinoma

**DOI:** 10.3390/cancers14194715

**Published:** 2022-09-28

**Authors:** Xingxing Gao, Hechen Huang, Caixu Pan, Zhibin Mei, Shengyong Yin, Lin Zhou, Shusen Zheng

**Affiliations:** 1Division of Hepatobiliary and Pancreatic Surgery, Department of Surgery, The First Affiliated Hospital, Zhejiang University School of Medicine, Hangzhou 310003, China; 2NHC Key Laboratory of Combined Multi-Organ Transplantation, Hangzhou 310003, China; 3Key Laboratory of the Diagnosis and Treatment of Organ Transplantation, Research Unit of Collaborative Diagnosis and Treatment for Hepatobiliary and Pancreatic Cancer, Chinese Academy of Medical Sciences, Hangzhou 310003, China; 4Key Laboratory of Organ Transplantation, Research Center for Diagnosis and Treatment of Hepatobiliary Diseases, Hangzhou 310003, China

**Keywords:** disulfiram, immunogenic cell death, NPL4, immunotherapy, CD47 blockade

## Abstract

**Simple Summary:**

The combination of disulfiram and copper is a novel anti-cancer drug under clinical development for the treatment of several tumor types including hepatocellular carcinoma. In this study, we investigated the potential of disulfiram and copper to induce immunogenic cell death and whether it could enhance the efficacy of immune checkpoint blockade. Our results showed that treatment with disulfiram and copper induced the release of damage-associated molecular patterns, such as calreticulin, ATP, and high mobility group box 1; thus, eliciting the maturation and activation of dendritic cells. The treatment with disulfiram and copper further enhanced the efficacy of CD47 blockade. Mechanically, disulfiram and copper promoted the nuclear accumulation and aggregation of nuclear protein localization protein 4 to inhibit the ubiquitin-proteasome system, thus, inducing endoplasmic reticulum stress. Taken together, the present findings suggest the potential clinical applications of disulfiram and copper in hepatocellular carcinoma.

**Abstract:**

Some chemotherapeutic agents have been found to enhance antitumor immunity by inducing immunogenic cell death (ICD). The combination of disulfiram (DSF) and copper (Cu) has demonstrated anti-tumor effects in a range of malignancies including hepatocellular carcinoma (HCC). However, the potential of DSF/Cu as an ICD inducer and whether it can enhance the efficacy of the immune checkpoint blockade in HCC remains unknown. Here, we showed that DSF/Cu-treated HCC cells exhibited characteristics of ICD in vitro, such as calreticulin (CRT) exposure, ATP secretion, and high mobility group box 1 (HMGB1) release. DSF/Cu-treated HCC cells elicited significant immune memory in a vaccination assay. DSF/Cu treatment promoted dendritic cell activation and maturation. The combination of DSF/Cu and CD47 blockade further facilitated DC maturation and subsequently enhanced CD8^+^ T cell cytotoxicity. Mechanically, DSF/Cu promoted the nuclear accumulation and aggregation of nuclear protein localization protein 4 (NPL4) to inhibit the ubiquitin-proteasome system; thus, inducing endoplasmic reticulum (ER) stress. The inhibition of NPL4 induced ICD-associated damage-associated molecular patterns. Collectively, our findings demonstrated that DSF/Cu-induced ICD-mediated immune activation in HCC enhanced the efficacy of CD47 blockade.

## 1. Introduction

Hepatocellular carcinoma (HCC), one of the most common malignant tumors, is the predominant cause of cancer-related deaths and has limited treatment options [1]. The manipulation of the immune system has proven to be an effective strategy for cancer therapy, whereas the majority of HCC patients still experience limited benefits from immune-based therapy [2]. Cancer-induced immunotolerance in the tumor microenvironment may partially explain the failure of clinical outcomes [3]. As a primary metabolic organ, the liver evolves multiple immune-suppressive mechanisms by receiving non-self-antigens from nutrients or resident microbes [4]. The hepatic immunosuppressive environment is mainly maintained by Kupffer cells, regulatory T cells, and myeloid-derived suppressor cells, mediated by pro- and anti-inflammatory cytokines [5]. Therefore, the use of additional agents or specific treatments to convert the immune-suppressive microenvironment into an inflammatory microenvironment could be an efficient strategy in HCC therapy.

Immunogenic cell death (ICD) is a particular form of stress-dependent cell death that can drive an immune response against dead cell-related antigens coupled with the activation of cytotoxic T lymphocyte (CTL)-driven adaptive immunity [6,7]. ICD is characterized by the coordinated release of a series of damage-associated molecular patterns (DAMPs), including the exposure of calreticulin (CRT) on the surface of dying cancer cells, the secretion of adenosine triphosphate (ATP), the release of high mobility group box 1 (HMGB1) and the activation of type I interferon (IFN) response [8,9,10]. These DAMPs can recruit the dendritic cells (DCs) and function as ‘eat me’ signals and subsequently attract cytotoxic T lymphocytes into the tumor microenvironment [11]. Recently, some chemotherapeutic agents, including anthracyclines, proteasome inhibitors, and DNA-damaging agents, have been documented to elicit bona fide ICDs and achieve better efficacy in combination with immune checkpoint blockade [12,13,14,15,16]. Therefore, the induction of ICD by chemotherapeutic agents may provide a novel strategy to inhibit tumor progression by inducing broad antitumor immunity.

Disulfiram (DSF), an old anti-alcoholism drug, recently showed anti-tumor potential in several cancer types, including HCC, lung cancer, breast cancer, and glioma [17,18]. Researchers have demonstrated that DSF interacted with copper (Cu) to form the metabolite CuET, which acts as a shuttle for Cu to cross the cell membrane and release Cu under oxidative conditions; thus, targeting cancer via inhibiting the activity of p97-dependent proteasomes via nuclear protein localization protein 4 (NPL4) [19,20]. Despite this, the potential therapeutic applications of DSF/Cu in HCC need to be further explored. As a potential proteasome inhibitor, the potential of DSF/Cu to induce ICD also deserves further investigation. Sun and colleagues found that DSF/Cu could enhance the ionizing radiation (IR) induced ICD in breast cancer [21]. You and colleagues found that DSF/Cu treatment increases the membrane exposure of CRT [22]. Thus, we hypothesized that DSF/Cu induces ICD via the inhibition of a ubiquitin-proteasome system and could promote the antitumor effect of immunotherapy in HCC.

In the current study, we demonstrated that DSF/Cu induced ICD in HCC and subsequently activated DCs. DSF/Cu induced ER stress through the inhibition of the ubiquitin-proteasome system via the nuclear accumulation and aggregation of NPL4. DSF/Cu-mediated enhancement of tumor immunogenicity improved CD47 blockade efficacy in HCC.

## 2. Materials and Methods

### 2.1. Cell Culture

HCC-LM3, Huh7, and Hepa1–6 cells were obtained from America Type Culture Collection (Manassas, VA, USA) in 2017. The methods of cell culture are presented in Appendix A. 

### 2.2. Reagents

Disulfiram (Selleck, S1680) and copper gluconate (CuGlu, Sangon Biotech, A503014) with over 99% purity were used for the study. The antibody to NPL4 (sc-365796) was obtained from Santa Cruz. The antibodies to EIF2S1 (ab32157), XBP1 (ab220783), and EIF2S1 (phosphor S51) (ab32157) were obtained from Abcam. The antibodies to CHOP (A0221), Ubiquitin (A18185), β-tubulin (A7074), and Lamin-B (A19970) were obtained from Abclonal. The antibodies to Zombie NIR™ APC-CY7(423106), CD45 FITC (103108)/PE (103106), CD11c PE (117308)/APC (117310), I-Ab PE-CY7 (116420), CD80 PE-CY5.5 (104712), CD86 APC (105012), CD3 APC (100236), CD8 PE (100708)/APC-CY7 (100712), CD4 PE (130310), CD44 PE-CY7(103010), CD62L APC (104412), Granzyme B PE-CY7(372213), and IFN-γ PE-CY5.5(505821) were obtained from eBioscience (San Diego, CA, USA).

### 2.3. Cell Viability Assay

Procedures for the cell viability assay are described in Appendix A. 

### 2.4. Colony Formation Assays

Procedures for the colony formation assay are described in Appendix A. 

### 2.5. Determination of Apoptosis, Surface CRT, ATP and HMGB1 Release 

Cells were grown to 40–50% confluence in 6-well plates, washed, and then incubated with increasing concentrations of DSF/Cu for 24 h. Tumor cell death induced by DSF/Cu was assessed using an Apoptosis Detection Kit Annexin V/Propidium Iodide kit (Vazyme, Nanjing, China), and surface CRT (CST, 62304S, Danvers, MA, USA) was detected by flow cytometry. The supernatants of cells cultured under the conditions described above were evaluated for extracellular HMGB1 levels using an ELISA kit (Cusabio, CSB-E08223h and CSB-E08225m, Wuhan, China). Extracellular ATP was quantified using an Enhanced ATP Assay Kit (Beyotime, S0027, Shanghai, China).

### 2.6. Immunoblotting

Procedures for immunoblotting are described in Appendix A.

### 2.7. Immunofluorescence

Procedures for immunofluorescence are described in Appendix A. 

### 2.8. Tumor Model and Treatment

*Orthotopic Tumor model*: A 12.5 μL mixture of RPMI 1640 medium and Basement Membrane Matrix (with the ratio of 1:1, 354248, Corning, NY, USA) containing 10^5^ Hepa 1–6 cells was injected into the left liver lobe of mice to establish the orthotopic tumor-bearing model. According to Zdenek’s study, mice received oral administration of 0.15 mg kg^−1^ CuGlu and 50 mg kg^−1^ DSF every day from day 5 [19]. CuGlu was administered 2 h before DSF. For the CD47 blockade treatment, a monoclonal anti-mouse CD47 antibody (BE0270, BioXcell, Lebanon, NH, USA) was injected to the subcutaneous model (250 µg/mouse, i.p., on days 3, 6, 9, 12). At day 15, mice were euthanized, and the tumors were removed for measurement of tumor weight and size.

*Vaccination experiments*: 10^6^ dead Hepa1–6 cells treated with DSF/Cu were injected into the right flank of a male C57BL/6 mouse (4 weeks). After 7 days, 4 × 10^5^ live Hepa1–6 cells were injected into the left flank. Incidence and growth of tumors were monitored every 2–3 days. 

### 2.9. Dendritic Cell Activation and Phagocytosis Assays

Procedures for Dendritic cell activation and phagocytosis assays are described in Appendix A.

### 2.10. Flow Cytometry Staining

Procedures for flow cytometry staining are described in Appendix A. 

### 2.11. siRNA

Procedures for siRNA are described in Appendix A. 

### 2.12. RNA Extraction, RNA-seq and Data Analysis

LM3 cells were treated with DSF/Cu in a concentration of 0.2 μM for 24 h or transfected with siRNA-NC or NPL4 siRNA-1. Total RNA of normal control and treated cells was isolated by TRIzol (Life Technologies, Carlsbad, CA, USA). The mRNA was sequenced by Illumina HiSeq 2500. Clean data were deposited in the NCBI Gene Expression Omnibus (GSE211734/PRJNA871352) The sequencing reads were aligned to the human reference sequence (UCSC/hg38.p12) by HISAT2 [23]. The feature Counts function was performed for each gene count from trimmed reads against the GENCODE (release 30) transcript models [24]. Differential gene expression analysis was quantitated by edgeR [25]. The top 200 significantly differentially expressed genes were used for GO and KEGG enrichment. The analysis of differentially expressed genes and gene-set enrichment analysis was implemented by the R package of Cluster Profiler. 

### 2.13. Statistics 

The Student t test and Mann–Whitney U test were used to compare the statistical difference between pairs of groups using Prism GraphPad software (not significant (ns), *p* > 0.05; *, *p* < 0.05; **, *p* < 0.01; ***, *p* < 0.001). All the data were described as the mean ± SEM, unless otherwise stated. Flow cytometry data were analyzed by FlowJo.10 (TreeStar, San Francisco, CA, USA).

## 3. Results

### 3.1. The Combination of DSF/Cu Inhibits Cell Proliferation and Induces Apoptosis of HCC Cells

To evaluate the antitumor effect of DSF/Cu on HCC cell lines, we first treated LM3, Huh7, and Hepa1–6 cells with increasing concentrations of DSF and 1 μM Cu for 24 h. Results indicated that mono-treatment with DSF or Cu showed no obvious toxicity against HCC cell lines, whereas the combination of DSF and Cu exhibited strong extra cytotoxic activity against HCC cells dose-dependently (Figure 1A and Appendix A). Next, colony formation assays indicated that DSF/Cu markedly suppressed colony formation capacity in a concentration-dependent manner (Figure 1B). Consistently, the EdU assay implied that the proliferation of HCC cells was inhibited by DSF/Cu dose-dependently (Figure 1C). To further investigate whether the cytotoxicity of DSF/Cu is associated with apoptosis, the apoptotic phenotype of HCC cells treated with DSF/Cu was assessed. The results showed that the apoptotic proportion was significantly increased in a DSF concentration-dependent manner (Figure 1D and Appendix A). Collectively, these results revealed that the treatment with DSF/Cu can decrease malignant cell growth and induce apoptosis in HCC cells. 

### 3.2. DSF/Cu Induces ICD in HCC Cells and Enhances DC Activation 

ICD is a cell death modality that induces an immune response against dead-cell-related antigens, especially when they are released by cancer cells. Several biochemical markers of ICD have been identified, including the exposure of CRT on the cell membrane, the extracellular secretion of ATP and HMGB1, and the activation of the type I IFN response. We first evaluated the surface expression of CRT after 24 h of DSF/Cu treatment. We observed an increase in CRT on the cell surface of dying HCC cells in a DSF concentration-dependent manner (Figure 2A). HMGB1 is released extracellularly from dead tumor cells at the late stage of ICD. After 48 h of DSF/Cu treatment, we observed the increase in the HMGB1 levels in the cell supernatant (Figure 2B). Furthermore, the levels of extracellular ATP were also increased in HCC cells treated with DSF/Cu (Figure 2C). We further performed transcriptome sequencing of normal control and DSF/Cu-treated LM3 cells. GSEA analysis revealed the activation of type I IFN response and inflammatory response after the treatment with DSF/Cu (Figure 2D). The heatmap also showed that type I IFN response-associated genes were upregulated (Figure 2E). Overall, these results demonstrate that DSF/Cu induced ICD in HCC cell lines in a DSF concentration-dependent manner.

ICD-associated DAMPs released by cells undergoing ICD have been proven to regulate the function of specific innate immune cell subsets, such as DCs [12]. To determine whether the DSF/Cu-induced cell death is immunogenic, we co-cultured DSF/Cu-treated Hepa1–6 cells with mouse BMDCs. The results revealed an efficient DC phagocytosis of the treated Hepa1–6 cells (Figure 2F and Appendix A). The enhanced maturation of DCs was also estimated by cytometry, as shown by I-Ab, CD80, and CD86 surface expression (Figure 2G–I and Appendix A). Therefore, our findings suggested that DSF/Cu promotes DCs maturation and activation in HCC cells.

### 3.3. DSF/Cu Induces ER Stress through a Buildup of Poly-Ubiquitylated Proteins Mediated by NPL4 Aggregation in Cell-Nucleus 

To further explore the mechanism of DSF/Cu-induced ICD in HCC cells, we performed functional enrichment of differentially expressed genes between normal control and DSF/Cu-treated HCC cells. The most significantly expressed genes were enriched to protein ubiquitination, unfolded protein response, and ER stress (Figure 3A,B). ER stress plays a key role in the induction of ICD [26]. The phosphorylation of eIF2α, the initiation part of ER stress, has been proven to be a pathognomonic biomarker of ICD [27]. Thus, we next explored the level of eIF2α phosphorylation in HCC cells treated with DSF/Cu and observed an increased eIF2α phosphorylation at serine 51 (Figure 3C). In addition, X-box binding protein 1 splicing (XBP1s) and CHOP, the other component of ER stress, were also upregulated.

Unfolded protein response undergoes activation upon accumulation of misfolded proteins. Skrott and colleagues recently proposed that DSF/Cu inhibited the ubiquitin-proteasome system by inhibiting NPL4. We also observed an accumulation of poly-ubiquitylated proteins in HCC cells treated with increasing concentrations of DSF/Cu (Figure 3C). Western blot and immunofluorescence showed a nucleus aggregation and accumulation of NPL4 after DSF/Cu treatment (Figure 3D,E). According to the above results, we speculated that DSF/Cu-mediated aggregation of NPL4 led to the accumulation of unfolded proteins and ultimately ER stress.

### 3.4. Inhibition of NPL4 Induces ICD in Hepatocellular Carcinoma

To further explore whether the inhibition of NPL4 induces ICD in HCC cells, siRNAs were used to silence the expression of NPL4 in LM3 cells (Figure 4A). Indeed, inhibition of NPL4 led to a suppression of proliferation in LM3 cells (Figure 4B–D). In addition, NPL4 knockdown contributed to a higher proportion of apoptosis and induced upregulation of ICD markers, including CRT exposure, HMGB1 and ATP release (Figure 4E–H). Results of Western blot also confirmed the accumulation of poly-ubiquitylated proteins and ER stress (Figure 4I). We further performed a transcriptome of LM3 cells transfected with siRNA-NC or siRNA-NPL4. Functional analysis revealed the activation of the immune response, ER stress, and protein ubiquitination, which was consistent with cells treated with DSF/Cu (Figure 4J). GSEA analysis and heatmap confirmed the activation of type I interferon response and interferon-gamma response (Figure 4K–L). Altogether, these data demonstrated that inhibition of NPL4 could induce ICD in HCC cells which were similar to DSF/Cu treatment.

### 3.5. DSF/Cu induced Immune Memory and DC Activation In Vivo

We next validated the immunogenic potential of DSF/Cu in a vaccine setting (Figure 5A). Our results revealed significant inhibition of tumor growth and enhanced tumor-free survival among mice injected with DSF/Cu-treated Hepa1–6 cells compared with negative controls (injected with PBS) (Figure 5B–D). Meanwhile, we evaluated the proportion of memory T cells in the lymph nodes at the contralateral flank. The proportion of effector memory CD4^+^ and CD8^+^ T cells (CD44^−^, CD62L^+^) was higher compared with negative controls (Figure 5E). The percentage of central memory T cells (CD44^+^, CD62L^+^) showed no significance between the two groups (Figure 5E). These results suggested that Hepa1–6 cells treated with DSF/Cu mediated the generation of immune memory. We also constructed an orthotopic tumor-bearing mouse model to address the capacity of DSF/Cu to inhibit tumor growth (Figure 5F). The results showed that DSF/Cu significantly inhibited the tumor growth (Figure 5G–I). Flow cytometry analysis of tumors showed that DSF/Cu treatment significantly increased the infiltration of DC cells expressing I-Ab and the costimulatory molecule (Figure 5J–L). Taken together, our results indicated that DSF/Cu induced the anti-tumor immune response in vivo.

### 3.6. Combination of DSF/Cu and Anti-CD47 Therapy Exhibited Great Antitumor Activity

CD47 blockade has emerged as a promising strategy for cancer immunotherapy [28]. A previous study has suggested that the expression of CRT promoted anti-CD47 antibody-mediated phagocytosis [29]. Given that DSF/Cu induced the membrane exposure of CRT, we further investigated whether DSF/Cu acts synergistically with anti-CD47 through the induction of ICD (Figure 6A). Our results showed that DSF/Cu alone or anti- CD47 alone could inhibit tumor growth, while the combination of two drugs had a more pronounced effect (Figure 6B–D). Compared with monotreatment with DSF/Cu or CD47 blockade, the frequencies of I-Ab^+^, CD80^+^ and CD86^+^DC cells were increased in the combination-treated group (Figure 6E–G). Moreover, secretion of IFN-γ and Granzyme B in infiltrated CD8^+^ T cells in the tumor microenvironment was also increased (Figure 6 H–I). Therefore, the above data suggested that the combination therapy of the DSF/Cu and CD47 blockade could effectively activate anti-tumor immunity to inhibit tumor growth.

## 4. Discussion

The development of novel drugs for selectively activating the ICD pathway holds great promise for cancer treatment. Recently, several anticancer agents that induce ICD have been approved for clinical application, including belantamab mafodotin and lurbinectedin, suggesting the broad prospects of ICD in cancer therapy [30,31]. DSF/Cu has shown anti-tumor effects in a variety of tumors, while the potential of DSF/Cu to induce ICD in HCC has not been reported. Here, we found that DSF/Cu induced CRT exposure, HMGB1 release, ATP secretion, and IFN pathway activation in HCC cells. These DAMPs-releasing cells further promoted DC maturation and activation. The tumor vaccination assays validated that the DSF/Cu-induced ICD elicits immune memory. In vivo experiments further confirmed that the oral administration of DSF/Cu inhibited tumor growth. DSF/Cu treatment increased mature CD11c^+^ DCs in the tumor microenvironment. Mechanistically, DSF/Cu indirectly inhibits the function of the proteasome by promoting the nuclear accumulation of p97 segregase adaptor NPL4, thereby activating the unfolded protein response (Figure 7). Collectively, our study indicated the potential of DSF/Cu as a promising therapeutic agent against HCC cells through ICD.

NPL4, one of the most versatile cofactors of p97, is involved in more than half of the p97-mediated cellular processes, such as endoplasmic reticulum-associated degradation (ERAD) and cell death [32]. During ERAD, NPL4 is recruited by p97 to extract polyubiquitinated proteins from the ER membrane, followed by processing and delivering the extracted proteins to the proteasome for degradation [33]. Intranuclear aggregation of NPL4 mediated by DSF/Cu treatment may prevent the trafficking of misfolded proteins to the proteasome, thereby inducing unfolded protein responses and ultimately triggering ICD. Hyperactivation of the ubiquitin-proteasome system is an important factor in abnormal tumor proliferation. Upregulation of NPL4 is observed in many cancer types and was associated with poor prognosis in HCC, melanoma, kidney chromophobe, and glioma (Appendix A) [34,35,36]. Therefore, the inhibition of p97/NPL4 segregase also represents an attractive approach for cancer therapy. Our study demonstrated that knockdown of NPL4 in HCC cells inhibited cell proliferation and promoted cell death, indicating that the DSF/Cu-induced ICD is mediated by NPL4 as the relevant target.

Intuitively, the combination of immunotherapy and ICD-inducing agents represents a potential therapeutic strategy, particularly in tumors lacking an effective immune response. The interactions between ICD-inducing chemotherapeutic drugs and the host’s immune system have been found to enhance the efficacy of the checkpoint blockade. A variety of ongoing clinical trials are assessing the effectiveness of the combination of checkpoint blockade and ICD inducers, which may benefit numerous patients and broaden the indications for treatments with existing chemotherapeutic agents, thereby reducing the side effects by dose reduction. In recent years, much progress has been made in targeting CD47 for cancer immunotherapy in solid tumors [37]. ICD inducers can promote the expression of pro-phagocytic signals in tumor cells, while anti-CD47 treatment can block the “don’t eat me” signal. The combined application of these agents may mediate the phagocytosis of tumor cells by phagocytes. Zhou and colleagues found that CD47 blockade and ICD induction efficiently boosted antitumor immunity and inhibited the tumor growth [38]. Our results further showed that DSF/Cu and anti-CD47 combination therapy enhanced the activation of DCs and CD8^+^ T cells in the tumor microenvironment and improved antitumor response.

In recent years, several new treatments for unresectable liver cancer have been developed, for instance, radiofrequency ablation, a multikinase inhibitor, and immune checkpoint blockade [39,40]. Immune checkpoint blockade has seen a fast development in the treatment of liver cancer during this time. As the monotherapy of immune checkpoint blockade showed non-statistically significant benefits in several clinical trials, researchers are exploring combination strategies of immunotherapy and other anti-cancer agents [41,42]. Several combination strategies, such as atezolizumab plus bevacizumab and lenvatinib plus pembrolizumab, have been demonstrated to better improve the prognosis of HCC patients [43,44]. Some previous studies and our study found DSF/Cu could activate the tumor immune microenvironment in preclinical models, indicating that the combination of DSF/Cu and immune checkpoint blockade may become a new treatment option for surgically unresectable liver cancer. 

## 5. Conclusions

In conclusion, this study demonstrated that DSF/Cu induced ICD in hepatocellular carcinoma by the inhibition of p97/NPL4 segregase. Our study further provides evidence that the anti-tumor effects of DSF/Cu are partially mediated by the activation of innate immunity via ICD. The combination of DSF/Cu and CD47 blockades might provide a new strategy for the improvement of immunotherapy.

## Figures and Tables

**Figure 1 cancers-14-04715-f001:**
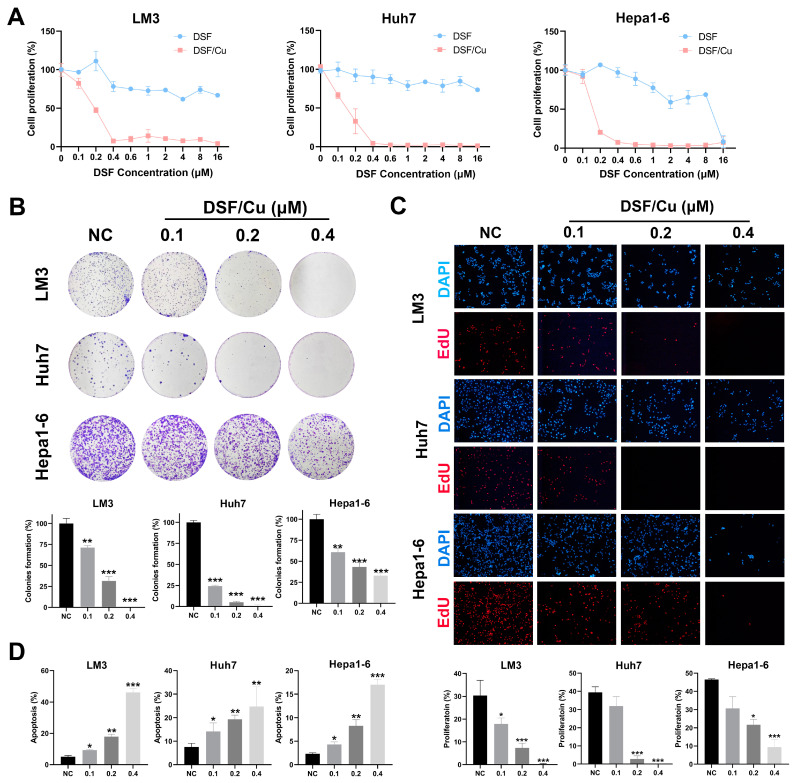
DSF/Cu inhibits proliferation and induces cell death in HCC cells. (**A**) Cell viability of LM3, Huh7, and Hepa1–6 treated with DSF/Cu at indicated concentration for 24 h. (**B**) The colony formation of LM3, Huh7, and Hepa1–6 cells under the treatment of DSF/Cu at indicated concentration. (**C**) The proliferation of LM3, Huh7, and Hepa1–6 HCC cells treated with DSF/Cu at indicated concentration were detected by Edu assay. Images with magnification at 100× are shown here. (**D**) Apoptosis analysis of LM3, Huh7, and Hepa1–6 cells treated with indicated concentration of DSF/Cu. *, *p* < 0.05; **, *p* < 0.01; ***, *p* < 0.001.

**Figure 2 cancers-14-04715-f002:**
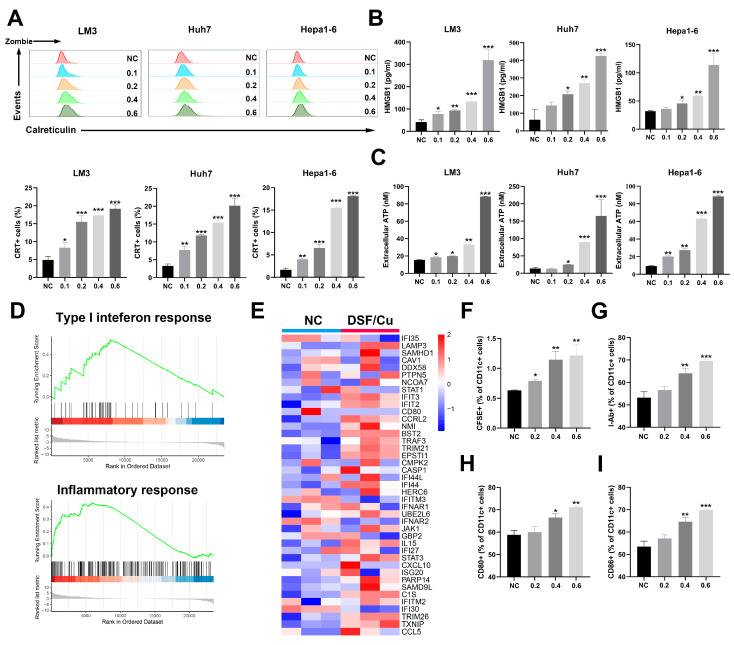
DSF/Cu induces ICD and promotes DC cell activation in HCC. (**A**) Membrane exposure of CRT on HCC cells after treatment with DSF/Cu was detected by flow cytometry. (**B**) Soluble HMGB1 in the media of HCC cells treated with DSF/Cu at different concentrations. (**C**) Extracellular ATP in the media of HCC treated with DSF/Cu at indicated concentration. (**D**) Gene-set enrichment analysis of type I interferon response- and inflammatory response-related genes. (**E**) Heatmap of type I interferon response-related genes. (**F**–**I**) The expression of CFSE, I-Ab, CD80, and CD86 in CD11c^+^ DC cells cocultured with Hepa1–6 cells treated with DSF/Cu was detected by flow cytometry. *, *p* < 0.05; **, *p* < 0.01; ***, *p* < 0.001.

**Figure 3 cancers-14-04715-f003:**
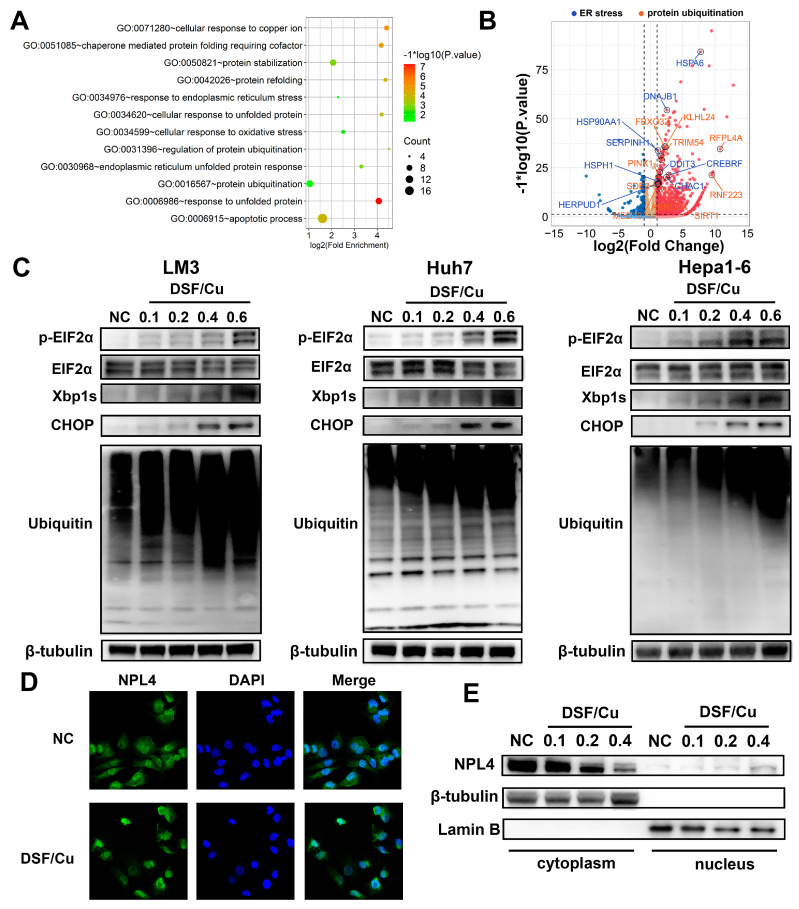
DSF/Cu induces unfolded protein response and inhibits protein degradation. (**A**) Pathway analysis between normal control and DSF/Cu-treated HCC cells. (**B**) Differential expression of unfolded protein response- and protein ubiquitination-related genes between normal control and DSF/Cu-treated HCC cells. (**C**) Western blot analysis of ER stress and poly-ubiquitylated proteins in LM3, Huh7, and Hepa1–6 cells after the treatment of DSF/Cu. (**D**) Fluorescence imaging of NPL4, cell nuclei were stained by DAPI. Images with magnification at 800× are shown here. (**E**) Western blot analysis of cytoplasmic and nuclear NPL4 in DSF/Cu-treated LM3 cells. The uncropped blots are shown in Appendix A.

**Figure 4 cancers-14-04715-f004:**
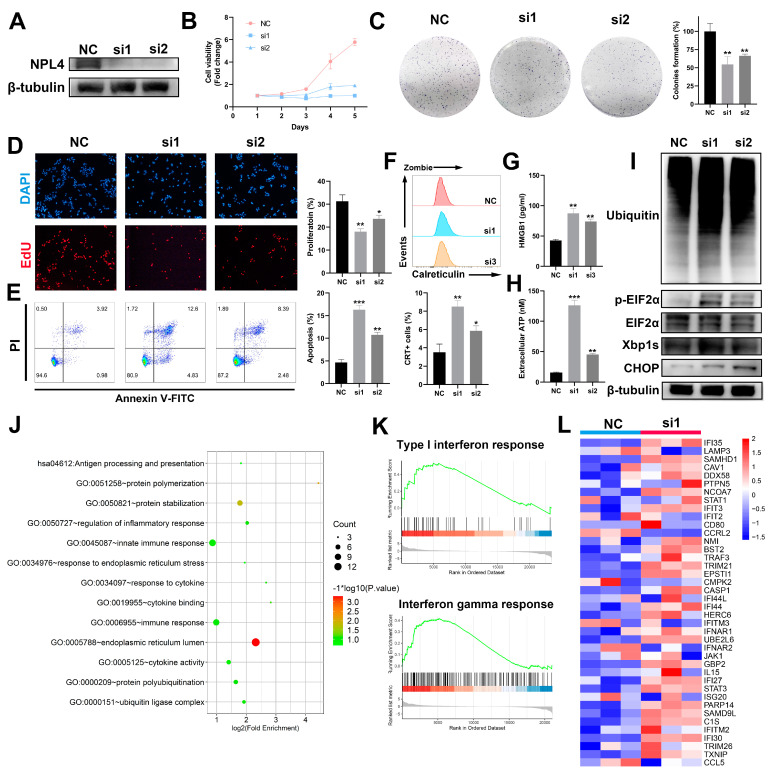
Inhibition of NPL4 inhibits proliferation and induces ICD in HCC cells. (**A**) Inhibition efficiency of NPL4 in LM3 cells through Western blot analysis. (**B**) The role of NPL4 inhibition in cell viability was detected by CCK-8 assay. (**C**) The role of NPL4 inhibition in clone-forming capacity of LM3 cells. (**D**) The role of NPL4 inhibition in cell proliferation was detected by Edu assay. Images with magnification at 100× are shown here. (**E**) Apoptosis analysis of LM3 cells after NPL4 inhibition. (**F**) CRT exposure, (**G**) HMGB1 release, and (**H**) ATP secretion of LM3 cells after NPL4 inhibition. (**I**) Western blot analysis of unfolded protein response and poly-ubiquitylated proteins in LM3 cells after NPL4 inhibition. (**J**) Pathway analysis of LM3 cells after NPL4 inhibition. (**K**) Gene-set enrichment analysis of type Ⅰ interferon response- and interferon-gamma response-related genes in LM3 cells after NPL4 inhibition. (**L**) Heatmap of type Ⅰ interferon response-related genes in LM3 cells after NPL4 inhibition. *: *p* < 0.05; **, *p* < 0.01; ***, *p* < 0.001. The uncropped blots are shown in Appendix A.

**Figure 5 cancers-14-04715-f005:**
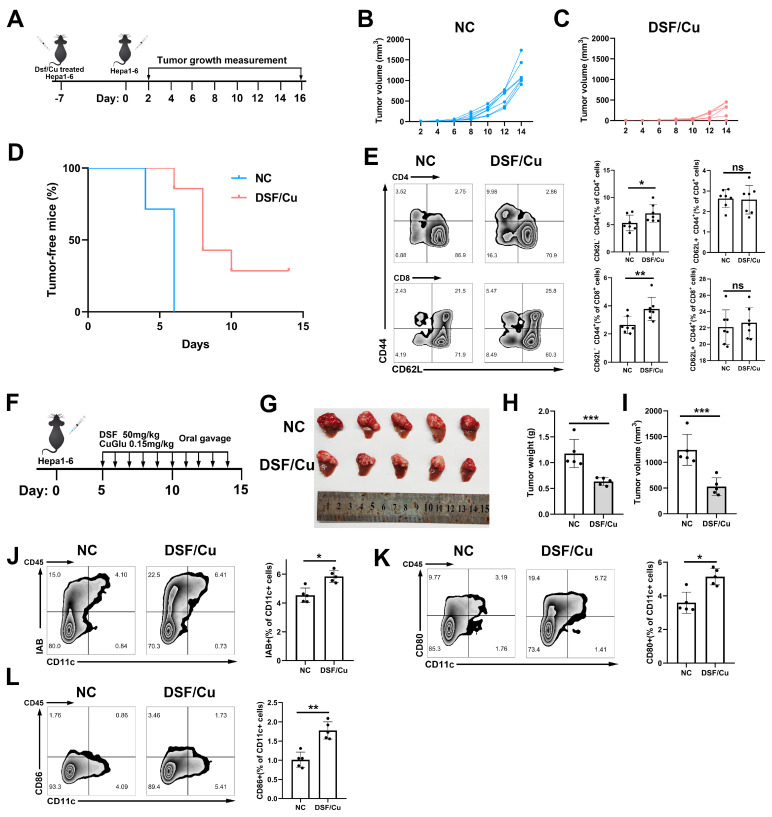
DSF/Cu induced intratumor immune cell infiltration in vivo. (**A**) A schematic view of the vaccination experiments. Individual tumor growth of normal control (**B**) or DSF/Cu-treated Hepa1–6 cells (**C**) measured every 2 days (n = 7 for each group). (**D**) Tumor-free survival analysis by Kaplan–Meier curves. (**E**) Analysis of the percentage of effector memory- (CD44^−^, CD62L^+^) and central memory- (CD44^+^, CD62L^+^) T cells. (**F**) A schematic view of the orthotopic tumor model for DSF/Cu treatment. (**G**) Tumor appearance, (**H**) weight and (**I**) volume were compared with the other group (n = 5). Analysis of the percentage of (**J**) I-Ab^+^, (**K**) CD80^+^, (**L**) CD86^+^ DC cells in CD45^+^ cells. ns: *p* > 0.05; *: *p* < 0.05; **: *p* < 0.01; ***: *p* < 0.001.

**Figure 6 cancers-14-04715-f006:**
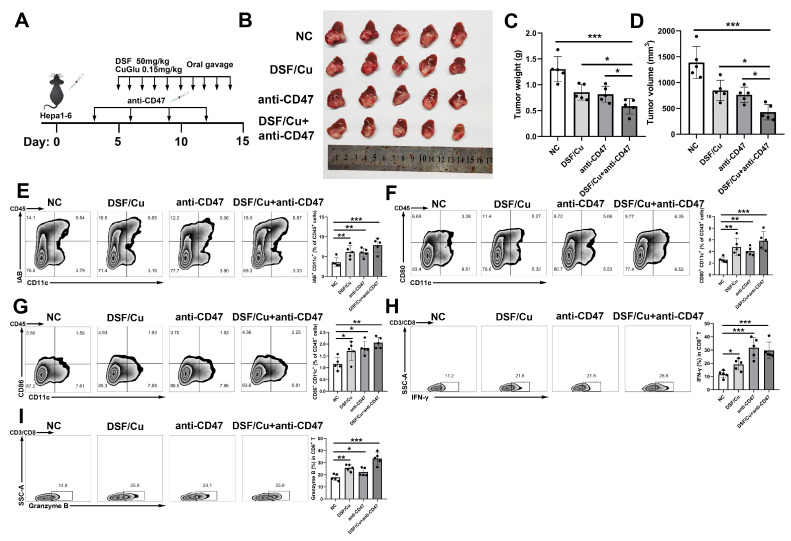
DSF/Cu enhances CD47 blockade activities in vivo. (**A**) A schematic view of the orthotopic tumor model for DSF/Cu and anti-CD47 treatment. (**B**) Tumor appearance, (**C**) weight and (**D**) volume were compared with each group (n = 5). Analysis of the percentage of (**E**) I-Ab^+^, (**F**) CD80^+^, (**G**) CD86^+^ DC cells in CD45^+^ cells. Analysis of the percentage of IFN-γ (**H**) and Granzyme B (**I**) in CD8^+^ T cells. ns: *p* > 0.05; *: *p* < 0.05; **: *p* < 0.01; ***: *p* < 0.001.

**Figure 7 cancers-14-04715-f007:**
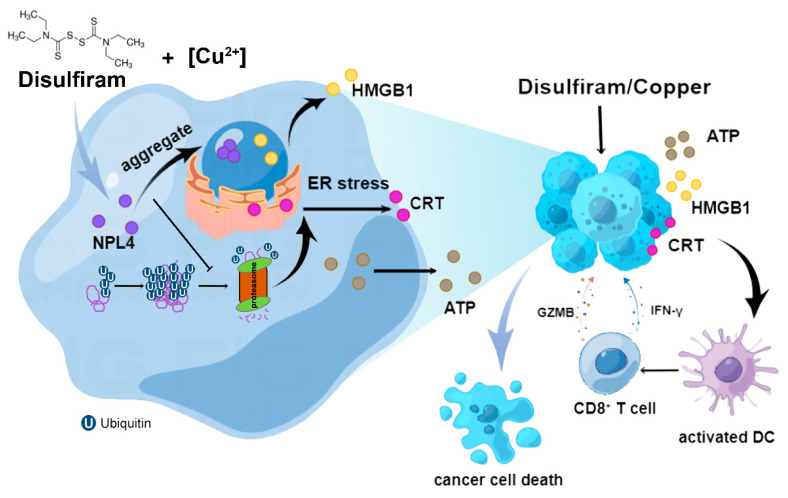
Schematic diagram showing that DSF/Cu induced ICD and activated antitumor immunity in HCC.

## Data Availability

All data needed to evaluate the conclusions in the paper are presented in the paper and/or the Appendix A. Additional data related to this paper may requested from authors.

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
