# Peer review of "Disulfiram/Copper Induces Immunogenic Cell Death and Enhances CD47 Blockade in Hepatocellular Carcinoma"

_cancers, 2022, doi:10.3390/cancers14194715_

Round 1
Reviewer 1 Report
Xingxing Gao et al., worked on DSF/Cu to claim the antitumor efficiency with the title of “Disulfiram/copper induces immunogenic cell death and enhances CD47 blockade in hepatocellular carcinoma”. I appreciate all the authors and their consistent work to claim the toxic effect of DSF/C combination. Authors confirmed the inhibition of p97/NPL4 segregase with all aspects.
Authors are requested to resolve some queries.
With few suggestions, I would recommend authors to modify their manuscript by following
1. Experimental section, 2.8. Tumor model and treatment: This section, authors can be rewrite elaborately to make a clear idea of the method and to follow researchers in future including
1. Did authors measured tumor size?
2. After how long over the period of tumor confirmation, authors started to oral administration of DSF/CuGlu?
3. Since the trial phase, Glioblastoma patients were administered with 80mg/1.5 mg of DSF and Cu (https://clinicaltrials.gov/ct2/show/NCT03034135). On what basis the concentration has been fixed for animal (if any good reference is available, can be included).
2. When authors used DSF and Cu separate compound to evaluate the toxic effect in the cell line experiment, there was no significant effect. Subsequently, combined two of them shows excellent toxic effect.
Question is, If each compound should have minimum inhibitory effect then only extra efficient activity will be explore when it undergoes together. There is contradiction over here.
If authors explain the reason or any reference to prove this could be easy to understand for the readers.
Author Response
Comment #1: Experimental section, 2.8. Tumor model and treatment: This section, authors can be rewrite elaborately to make a clear idea of the method and to follow researchers in future including
- Did authors measured tumor size?
- After how long over the period of tumor confirmation, authors started to oral administration of DSF/CuGlu?
- Since the trial phase, Glioblastoma patients were administered with 80mg/1.5 mg of DSF and Cu (https://clinicaltrials.gov/ct2/show/NCT03034135). On what basis the concentration has been fixed for animal (if any good reference is available, can be included).
Response: We are sincerely grateful for this suggestion. For your questions, we will answer point-to-point.
- Thanks for the concerning. We have added a description of measuring mouse tumor mass and volume in section, 2.8. Tumor model and treatment (Page 3, line 133). The tumor sizes of Orthotopic Tumor model were showed in Figure 5I and Figure 6D.
- We are very sorry for this confusion. According to Zdenek Skrott’s research (PMID: 29211715), they started the treatment of DSF/Cu when tumor grew to 100 mm3. In our pilot experiments, the tumor volume in mice reached 100 mm3 in 5-7 days (Figure S1). Based on this result, we chose to start DSF/Cu treatment on day 5 after model bearing. We have added a description of when to start the treatment of DSF/Cu in section, 2.8. Tumor model and treatment (Page 3, line 130).
- Thanks for the concerning. According to Zdenek Skrott’s research (PMID: 29211715), they selected the concentration of 50mg/1.5 mg of DSF and CuGlu in mouse model. Xueying Ren and colleagues (PMID: 34482117) selected the concentration of 50mg/0.6 mg of DSF and CuCl2 (intramuscular injection) in a mouse model. Taking into account their experimental designs, we chose this experimental protocol (Page 3, line 129).
Comment #2: When authors used DSF and Cu separate compound to evaluate the toxic effect in the cell line experiment, there was no significant effect. Subsequently, combined two of them shows excellent toxic effect. Question is, if each compound should have minimum inhibitory effect then only extra efficient activity will be explore when it undergoes together. There is contradiction over here.
Response: We greatly appreciate this comment that would make a more scientific manuscript. We have modified the description of the results in the manuscript accordingly (Page 4, line 168). As previous studies reported (PMID: 29211715; PMID: 33402676), DSF interacted with Cu to form the metabolite CuET, which acts as a shuttle for Cu to cross the cell membrane release Cu under oxidative conditions, thus inhibiting the activity of NPL4. This might explain the extra cytotoxic effects of tumor cells under the combination treatment of DSF and Cu. We have added the potential mechanism of the extra cytotoxic activity on cancer cells under the combination of DSF/Cu in the Introduction section (Page 2, line 80).
Reviewer 2 Report
This preclinical study assesses a current, timely topic in HCC and the authors are to be commended for bringing this interesting experience to this field.
We recommend some changes:
- A linguistic revision by a professional service is recommended since there are some grammar mistakes and oversights to be corrected
- A more personal perspective discussing how this study could have an impact on everyday clinical practice should be included
- The background of the changing scenario of medical treatment in HCC should be better discussed, and some recent papers regarding this topic should be included in the introduction section (PMID: 34431725 ; PMID: 32684988 ), for a matter of consistency. In addition, more recent references should be included, since quite old papers are reported by the authors.
Major changes are necessary.
Author Response
Comment #1: A linguistic revision by a professional service is recommended since there are some grammar mistakes and oversights to be corrected
Response: Thanks for your suggestion. We have addressed several language and grammer mistakes in our manuscript. We have also corrected the English version of this article at https://www.mdpi.com/authors/english.
Comment #2: A more personal perspective discussing how this study could have an impact on everyday clinical practice should be included.
Response: We greatly appreciate this comment that would make a more scientific manuscript. We have added a paragraph in Discussion section to discuss the impact of our study on clinical practice of HCC (Page 14, line 374).
Comment #3: The background of the changing scenario of medical treatment in HCC should be better discussed, and some recent papers regarding this topic should be included in the introduction section (PMID: 34431725 ; PMID: 32684988 ), for a matter of consistency. In addition, more recent references should be included, since quite old papers are reported by the authors.
Response: Thanks for your suggestion. We have added the description of background of the changing scenario of medical treatment in HCC in the Discussion section (Page 14, line 366). We are appreciated for your recommendation. We have cited the 2 papers (PMID: 34431725; PMID: 32684988) you recommended in the discussion chapter, which were of great benefit to improve the quality of our study. Furthermore, we have replaced some old papers with newer ones.
Round 2
Reviewer 2 Report
Acceptance.